# Plasmon-Enhanced Photothermal and Optomechanical Deformations of a Gold Nanoparticle

**DOI:** 10.3390/nano10091881

**Published:** 2020-09-20

**Authors:** Jiunn-Woei Liaw, Guanting Liu, Yun-Cheng Ku, Mao-Kuen Kuo

**Affiliations:** 1Department of Mechanical Engineering, Chang Gung University, Taoyuan 33302, Taiwan; 2Department of Mechanical Engineering, Ming Chi University of Technology, New Taipei City 243303, Taiwan; 3Medical Physics Research Center, Institute for Radiological Research, Chang Gung University and Chang Gung Memorial Hospital, Taoyuan 33302, Taiwan; 4Center for Advanced Molecular Imaging and Translation, Chang Gung Memorial Hospital, Taoyuan 33305, Taiwan; 5Institute of Applied Mechanics, National Taiwan University, Taipei 10617, Taiwan; sax8529567@gmail.com (G.L.); d05543005@ntu.edu.tw (Y.-C.K.)

**Keywords:** photothermal, optomechanical, plasmon, gold nanoparticle, nanorod, dimer, Maxwell’s stress, surface tension, traction

## Abstract

Plasmon-enhanced photothermal and optomechanical effects on deforming and reshaping a gold nanoparticle (NP) are studied theoretically. A previous paper (Wang and Ding, ACS Nano 13, 32–37, 2019) has shown that a spherical gold nanoparticle (NP) irradiated by a tightly focused laser beam can be deformed into an elongated nanorod (NR) and even chopped in half (a dimer). The mechanism is supposed to be caused by photothermal heating for softening NP associated with optical traction for follow-up deformation. In this paper, our study focuses on deformation induced by Maxwell’s stress provided by a linearly polarized Gaussian beam upon the surface of a thermal-softened NP/NR. We use an elastic model to numerically calculate deformation according to optical traction and a viscoelastic model to theoretically estimate the following creep (elongation) as temperature nears the melting point. Our results indicate that a stretching traction at the two ends of the NP/NR causes elongation and a pinching traction at the middle causes a dent. Hence, a bigger NP can be elongated and then cut into two pieces (a dimer) at the dent due to the optomechanical effect. As the continuous heating process induces premelting of NPs, a quasi-liquid layer is formed first and then an outer liquid layer is induced due to reduction of surface energy, which was predicted by previous works of molecular dynamics simulation. Subsequently, we use the Young–Laplace model to investigate the surface tension effect on the following deformation. This study may provide an insight into utilizing the photothermal effect associated with optomechanical manipulation to tailor gold nanostructures.

## 1. Introduction

Studies of light–matter interaction in physics and chemistry at the nanoscale have attracted lots of attention for decades [1,2,3,4,5]. Recently, interesting experimental results [1] have shown that a spherical gold nanoparticle (NP) can be deformed into an elongated NR or even a dimer (two separated NPs) as irradiated by a tightly focused linearly polarized (LP) continuous-wave (CW) laser beam of 446 nm in air, where a P(VDF-TrFE) layer was coated on a Si substrate to reduce adhesion. In addition, twist deformation of a gold NP, irradiated by a circularly polarized laser beam, was reported in the Supporting Information of [1]. The mechanism could be due to the combination of the photothermal effect for softening the NP with optical stress (Maxwell’s stress) for the follow-up optomechanical deformation. Although the former causes softening of the NP without melting being confirmed, the role of the latter in deformation needs to be further identified. The photothermal effect on elongation of suspended gold NPs into NRs caused by 532-nm laser was also reported [2]. In addition, other research showed that optical stress can be used to bend a suspending gold nanorod (NR) in water by means of an LP laser beam of 1064 nm [6]. The mechanism of V-shaped deformation of an NR was theoretically explained in [7]; photothermal heating softens gold NR, and optical stress (Maxwell’s stress on surface) induces an optical bending on the NR by a LP Gaussian beam. In contrast, most previous experiments showed that a gold NR can be heated to become a spherical NP by irradiation of CW or pulsed lasers [8,9,10,11,12,13,14,15]. This is because as the power of laser is high enough to melt the NR the surface tension of liquid metal spherifies the NR [8]. Since the order of magnitude of surface tension effect is much larger than that of optical stress, the surface tension dominates the deformation of spherification if the melting of NP takes place. In addition, the bulging-belly deformation of gold NRs was also reported due to laser-induced partial softening and melting [16]. The mechanism of bulging belly, a swelling at the middle zone of an NR, might be attributed to shrinkage of surface tension of the liquid layer covering the surface of an NR with semi-solid state inside. The plasmonic photothermal effect of laser heating can also induce welding, soldering and sintering of multiple NPs [17,18,19,20,21]. As is well known, the melting point of gold NP could be much lower than 1064 °C, which is that of a bulk gold [22,23,24,25,26,27]. In general, the melting point of NP is size dependent: the smaller the size, the lower the melting temperature. A research reported that gold nanowires with a diameter of 30 to 50 nm start to fragment at 400 °C and become a chain of NPs as temperature is roughly higher than 600 °C [23]. Once the premelting of NP starts, the Rayleigh instability may happen [23,24,25,26,27]. When the temperature of the NP is lower than and close to 600 °C, it is in a semi-solid state and easily deformable; the NP behaves as a soft material with low Young’s and shear modulus [28]. However, heat transfer between NP and medium/substrate is not well understood, e.g., nanoscale thermal radiation [29]. Therefore, precise control of heating conditions (e.g., laser power) is very crucial to manipulate photothermal and optomechanical deformation of plasmonic nanostructures by means of light on demand. 

In this paper, we aim to study optomechanical deformation caused by Maxwell’s stress provided by an LP light upon the surface of a photothermal-softened gold NP/NR. The multiple multipole (MMP) method is utilized to simulate the electromagnetic field of a gold NP/NR irradiated by an LP Gaussian beam, and then to calculate the optical traction (Maxwell’s stress on surface) distribution on the surface of the NP/NR [7,30,31,32]. Next, an elastic model is used to numerically simulate the deformation according to the optical traction distribution, and a viscoelastic model is applied to theoretically estimate the following creep (elongation) as temperature nears the melting point [33,34]. In addition, we will also discuss the effect of quasi-liquid and liquid layers on deformation of a premelting NP/NR, which is predicted from the molecular dynamic simulation based on the reduction of surface energy [35,36]. We will adopt a classical model of Young–Laplace, using an equivalent surface tension on the surface (the effect of outer liquid layer) enclosed in a semi-solid material inside (an equivalent material of quasi-liquid and solid phases), to analyze the elastic deformation of NP before fully melting. Our analysis will illustrate how a gold NP can be stretched into an elongated NR with a neck and then divided into two NPs (a dimer) via the optical stress after being photothermal-softened [1]. 

## 2. Method

Figure 1a shows the different configurations of a gold NP/NR irradiated by a focused *x*-polarized Gaussian beam of 446 nm in air at six stages. The propagating direction of normally incident light is along the *z*-axis; wave vector is $k=\;-\;ke_{z}$. The waist of the Gaussian beam is $w_{0}=$ 300 nm. The fluence of laser beam is 25 MW/cm^2^. The multiple multipole (MMP) method is utilized to analyze the induced electromagnetic field [7,30,31,32]. The optical traction on the surface of the NP/NR is defined as the inner product of Maxwell’s stress on the surface with the normal vector, T·n. The Maxwell’s stress tensor (**T**) in terms of the electric and magnetic fields (**E**, **H**) on the surface of NP/NR is expressed by,
(1)T=12Re(εEE¯+μHH¯−12I(εE·E¯+μH·H¯))

In Equation (1), *ε* and *μ* are the permittivity and permeability of the surrounding medium (i.e., water); the overbar symbol and Re represent the complex conjugate and the real part, respectively. Due to the plasmonic photothermal effect, the irradiation of laser beam provides a heating power density in the NP,
(2)$$\rho_{V}=\frac{1}{2}\omega \varepsilon^{\prime \prime }E^{2}$$
which is proportional to the intensity of electric field. The permittivity (dielectric constant) of gold is expressed as $\varepsilon_{g}=\varepsilon^{\prime }+\varepsilon^{\prime \prime }$; the relative dielectric constant of gold at wavelength of 446 nm is (−1.7436, 5.3698) [37]. The heating power $P_{a}$ on NP/NR can be obtained by the volume integral of the heating power density,
(3)$$P_{a}=\int_{V}^{}\rho_{V}dV$$
where *V* is the volume of NP or NR. Next, FEM (COMSOL) is used to simulate the corresponding elastic deformation according to the applied optical traction. The material properties of Young’s modulus *E* and shear modulus *G* of gold are functions of temperature *T*. As temperature *T* of gold NP increases, *E* and *G* of NP are reduced due to thermal softening [28]. Since the Young’s modulus of gold nanowire is about *E_S_* = 78 GPa and the value of liquid gold is *E_L_* ≅ 0, it is difficult to estimate the Young’s module *E* in the transition zone of the melting point [38]. Although it should be a function of temperature, we assume that the Young’s modulus *E* of gold NP of semi-solid at the premelting stage (approximately 590 °C) is about *E* =$\;\sqrt{E_{S}E_{L}\;}$= 2 × 10^4^ Pa and E/G≈3 throughout this paper, where *E_L_* ≈ 0.005 Pa. Consequently, the softened NR becomes deformable as subjected to optical loading. In addition, viscoelastic deformation accompanies to exhibit the creep. According to the 1D Maxwell’s model for a viscoelastic material, the relation of strain $\varepsilon_{ve}$ and tensile stress σ for NP/NR can be roughly expressed as $\varepsilon_{ve}=\;\sigma \times \;(1/E\;+\;t/\eta )$; the former is due to elastic deformation and the latter is viscous creep. Here, we assume that the viscosity *η* is roughly $10^{2}$ Pa⋅sec, as the temperature is at the melting point (600 °C) [34]. If the tensile stress upon an NP provided by optical stress is about $\sigma =3\;\times \;10^{3}$ Pa, the incremental strain is 3.1 for *t* = 0.1 sec. This is to say that an NP can be deformed into an elongated NR with an aspect ratio of 3.1 via the photothermal and optomechanical effect.

Under continuous heating, the premelting of gold NP may take place. According to the estimate of molecular dynamic simulation, a quasi-liquid layer covering the entire surface of NP to reduce surface energy is induced at the onset of melting, and its thickness gradually increases toward the core as temperature rises [35,36]. Once the liquid layer is produced, the surface tension effect needs to be considered. Since the order of magnitude of surface tension effect is much larger than that of optical traction, the surface tension of liquid gold dominates the following deformation of NP. We use a classical model to calculate the following deformation; the Young–Laplace model of surface tension effect is used to simulate the deformation of a semi-solid NP subjected to an effective pressure on the surface,
(4)$$p=\gamma \left(\frac{1}{r_{1}}+\frac{1}{r_{2}}\right)$$
where $r_{1}$ and $r_{2}$ are the two principal radii of surface curvature of the NP. In general, the surface tension (energy) of a molten gold NP is size dependent [36]. Since the equivalent diameter of a single NP or a dimer (two NPs) in our analysis is as large as 80 nm, the surface tension (energy) *γ* adopted for calculation is about 1.45 J/m^2^ (N/m), according to the results of molecular dynamics simulation [21,36]. As a matter of fact, the spherification of gold nanostructures caused by surface tension of liquid gold is usually observed if a high-power laser beam is applied to overheat the nanostructures [8,9]. As shown in Figure 1a, the deformation and reshaping process of gold NP/NR is divided into six stages:

Stage I: A spherical NP is heated to be softened by laser beam irradiation. The optical traction stretches the NP to be an elongated NR [1,2].

Stage II and III: An elongated NR with a pronounced necking (a dumbbell with a dent at the center). A gradual creep of elongation occurs due to viscoelastic behavior of semi-solid gold as temperature nears the melting point (approximately 590 °C).

Stage IV, V and VI: If the laser irradiation continues, surface melting occurs at the center region to produce a NR dimer (two separated NRs) due to Rayleigh instability. Subsequently, the liquid layer covers the whole surface of each NR. Consequently, the surface tension makes them contractive to become two spherical NPs due to the reduction of surface energy. In addition, heating power is reduced because the plasmonic photothermal effect of liquid gold disappears. Subsequently, the liquid layer on the surfaces of the two NPs turns to semi-solid again. After the surface tension disappears, the optical stress dominates the follow-up deformation again. If the irradiation time is prolonged, optical tractions upon the two NPs will make them elongated. Hence, the spherical dimer will become an NR dimer once again. This interesting phenomenon has been observed in [1].

## 3. Numerical Results and Discussion

For the simulation of elastic deformation, we assume that there is a substrate supporting a gold NP/NR on the back side to avoid optical bending of the NP/NR, and the refractive index of substrate is the same with the surrounding medium to avoid the reflection of the incident light. In addition, the substrate effect on heat transfer is neglected in the following analysis. At the beginning, a gold NP of *r* = 40 nm is assumed to be at the center of the focal plane of a focused laser beam of 446 nm in air with a fluence of 25 MW/cm^2^. Although the deformation is a continuous process with incremental forming, we only discuss some typical states to demonstrate the photothermal effects associated with optomechanical deformation of the NP/NR. Throughout the following analysis, the volumes of different configurations are the same. The configurations of photothermally and optomechanically deformed gold NP irradiated by a LP Gaussian beam in air at six stages are shown in Figure 1a. The scattering cross section (SCS) spectra of the different configurations and the corresponding absorption (heating) power $P_{a}$ of each stage are plotted in Figure 1a,b, respectively. As pointed out by [1], we can monitor reconfiguration of the NP by measuring the change of SCS spectrum. As the shape of the NP becomes elongated (i.e., aspect ratio increases), the wavelength of the surface plasmon resonance (the peak of SCS spectrum) is red-shifted for Stage I to III. Nevertheless, when an elongated NR breaks into two NPs, the SCS spectrum turns back abruptly at Stage IV, but the surface plasmon resonance behavior of a dimer becomes stronger than that of a bigger single NP at Stage I. Herein, we would like to provide qualitative descriptions of the deformation rather than quantitative ones in the following. Through analysis of the MMP method, the intensity of electric field in the cross section of NP is shown in Figure 2a, and the optical traction distribution on the surface and the deformation are plotted in Figure 2b. For Stage I and II, the numerical result indicates that optical stretch at the two ends and optical pinch (pressure) at the upper center are induced due to optical traction. Based on optical traction distribution, the elastic deformation of NP is calculated using FEM (COMSOL), as shown in Figure 2b. A spherical shape is deformed to an oblate ellipsoid; the maximum displacement of deformation occurs at the upper central zone. This deformation is caused by irradiation of a downward propagating laser beam. Here, $E=2\times 10^{4}$ N/m^2^ and E/G≈3 of a softened gold NP are used for simulation. The order of magnitude of elastic strain is about 0.15. The longitudinal axis of the elongated NP is along the polarization of LP light. As *T* is at the onset of melting point, approximately 590 °C, the viscoelastic model estimates that a viscous creep causes an elongation of the deformed NP within 1 s, where the viscosity of semi-solid gold NP is roughly $10^{2}$ Pa⋅s.

As a result, an elongated gold NR (*r* = 25 nm with aspect ratio of 3.1), having the same volume with the original NP of *r* = 40 nm, is induced. The distributions of ${∥E∥}^{2}$ in the *x–z* cross section of NR at *y* = 0 and the optical traction on this NR under the same irradiation of laser beam are shown in Figure 3a,b, respectively. Again, the optical stretch at the two ends and optical pinch (pressure) at the center are observed in Figure 3b for Stage III. The elastic deformation of the NR is also plotted in Figure 3b, where the color represents the magnitude of displacement. The deformed NR becomes more flattened with a dent at the center and two obvious heads at the two ends; again, the maximum displacement of the deformation occurs at the upper central zone. This result can explain how the morphology of a dumbbell in the photothermal process is formed [1]. Again, the longitudinal axis of the elongated NR is assumed to be parallel to the polarization of LP light. In addition, the maximum heating power density occurs at the center part of NR, as shown in Figure 3a.

Subsequently, liquid gold is induced first at the center region of the NR due to higher local heating, and then Rayleigh instability causes the fragment of NR to become a NR dimer at Stage IV. We model a gold NR dimer of *r* = 29 nm with an aspect ratio of 1.21 and a gap of 14.7 nm for the simulation. Figure 4a shows the heating power density in the cross section of the *x–z* plane for Stage IV. The maximum heating power density occurs at the surfaces of the dimer near the gap zone due to plasmonic coupling. This process needs to take more heating time for the phase to change from solid to liquid; the latent heat of fusion for gold is about 63 kJ/kg. Once the liquid gold covers the surface, the Rayleigh instability occurs at the beginning of Stage IV; the NR becomes two discrete NRs. Subsequently, the surface tension continues to spherify the morphology with a minimum surface at Stage V and VI. The maximum displacement of the deformation occurs at the upper central zone with a swelling. Figure 4b illustrates that the morphology of the two NRs will be spherified by surface tension with *γ* of 1.45 J/m^2^ (N/m) [21,36]. An effective pressure of Young–Laplace model is applied on the surface of each semi-solid NR with Young’s modulus $E=2\times 10^{4}$ N/m^2^ and E/G≈3 inside. As the liquid layer increases, the plasmonic effect of NP/NR gradually disappears to reduce the heating power. However, the plasmon behavior of a transition solid-liquid phase of gold is too complicated to simulate. Although the permittivity (dielectric constant) of gold should be temperature dependent, we assume that the permittivity of gold is always the same for all simulation of the heating power in Figure 1b, and we consider the shape effect only. Our results demonstrate that heating power is gradually reduced from Stage IV to VI, as shown in Figure 1b. If this phenomenon happens, the liquid part will subsequently become solidified again after a period of time. Without surface tension, the optical traction dominates the optomechanical deformation again to stretch each NP of the dimer into elongated NR; the process goes round and round. The above analysis and explanation for the gradual deforming process of NP/NR are consistent with the observation of [1]. Since Wang and Ding’s group modified the Si substrate with a P(VDF-TrFE) layer for their experiment, the adhesion effect between a molten gold NP and the surface-modified substrate was reduced [1]. Hence, we did not take into account the adhesion effect in our simulation. However, if the surface of substrate is not modified, the adhesion effect may play an important role to suppress the optical-stretching deformation of a heated gold NP.

## 4. Conclusions

A variety of deformations and reshaping of gold NP/NR induced by plasmon-enhanced photothermal effects combined with optomechanical effects were studied theoretically in this paper; the former causes softening and the latter causes deformation of NP/NR. Through analysis of Maxwell’s stress upon the NP/NR, we found that a stretching traction at the two ends of a gold NP/NR causes elongation and a pinching traction at the middle part causes a dent; a higher local heating occurs at the central area. An elastic model was used to numerically calculate the deformation according to the optical traction, and a viscoelastic model was used to theoretically estimate the following viscous creep. Our results showed that the NP is deformed into an elongated NR with a dent; the schematic is shown in Figure 1. We also found that as the orientation of NR is parallel to the polarization of LP light the maximum heating power is induced to speed up the softening of NR. As the temperature of NP is very near the melting point, the thermal-softened gold NP/NR is semi-solid; it is simultaneously subjected to optical stress to deform as a soft material. However, if the laser power is increased or irradiation time is prolonged, the NR could start to melt. As a result, the surface tension of a quasi-liquid layer tends to minimize the surface to cause a spherical shape at the premelting stage; Rayleigh instability occurs [23,24,25,26]. Eventually, the elongated NR with a dent could be cut into two smaller NPs. Hence, it is critical to use laser-heating for reshaping a gold NP. In this study, we used multidisciplinary approaches to interpret the process of deformation and reshaping of gold NP/NR involving photothermal effects associated with the optomechanical mechanism. This study may provide an insight into utilizing photothermal effect associated with optomechanical manipulation to tailor the morphology of gold nanostructures by means of light.

## Figures and Tables

**Figure 1 nanomaterials-10-01881-f001:**
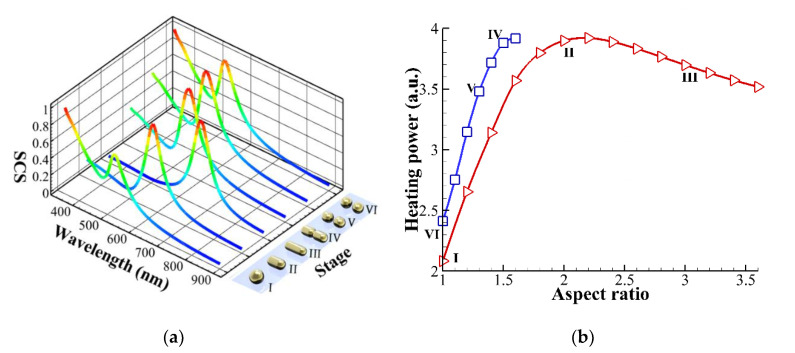
(**a**) Configurations of photothermally and optomechanically deformed gold nanoparticle (NP) of *r* = 40 nm at six stages, as irradiated by a linearly polarized (LP) Gaussian beam of 446 nm with a waist of 300 nm in air. The scattering cross section (SCS) spectra of different configurations with the same volume at these stages are plotted. (**b**) Total heating power versus aspect ratio for different configurations with the same volume. Line with triangle: a single NP or nanorod (NR), and line with square: an NR/NP dimer with different aspect ratio.

**Figure 2 nanomaterials-10-01881-f002:**
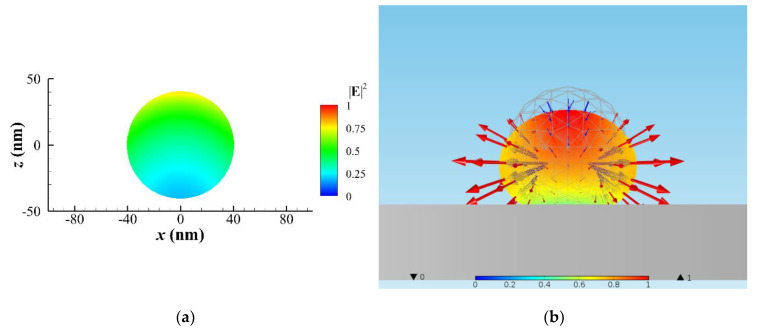
A spherical NP of *r* = 40 nm irradiated by an *x*-polarized Gaussian beam of 446 nm in air. (**a**) ${∥E∥}^{2}$ distribution in the *x–z* cross section of NP at *y* = 0. (**b**) The optical traction distribution (vector: optical traction) on the surface of the NP, and the morphology of the elastically deformed NP (color: magnitude of normalized displacement) for Stage I and II, where the inward arrows (blue): pressing traction (pressure) and the outward arrows (red): stretching traction. A spherical NP is deformed to an oblate-ellipsoidal one.

**Figure 3 nanomaterials-10-01881-f003:**
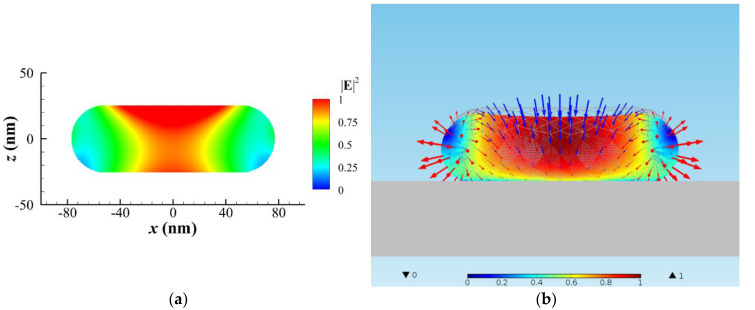
A typical NR of *r* = 25 nm with an aspect ratio of 3.1 irradiated by a Gaussian beam of 446 nm in air for Stage III. (**a**) ${∥E∥}^{2}$ distribution in the *x–z* cross section of NR at *y* = 0. (**b**) The optical traction distribution (vector: traction) on the surface of NR, and the morphology of the elastically deformed NR (color: magnitude of normalized displacement), where the inward arrows (blue): pressing traction (pressure) and the outward arrows (red): stretching traction. A dent is caused by the optical traction at the upper central zone of the elongated NR.

**Figure 4 nanomaterials-10-01881-f004:**
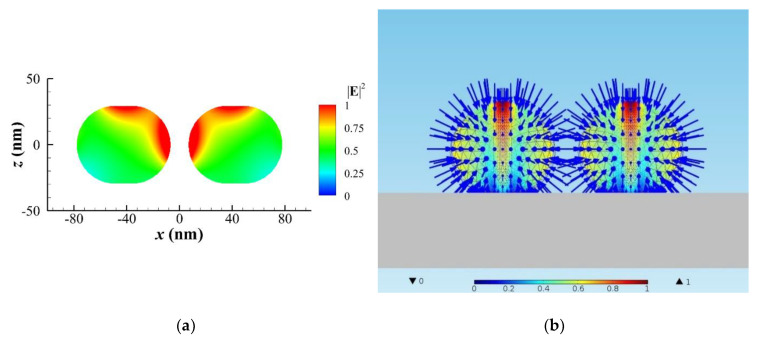
A gold NR dimer of *r* = 29 nm with an aspect ratio of 1.21 and a gap of 14.7 nm irradiated by an *x*-polarized Gaussian beam of 446 nm in air for Stage IV to VI. (**a**) ${∥E∥}^{2}$ distribution in the *x–z* cross section of NR dimer at *y* = 0. (**b**) The spherification of each elastically deformed NR was subjected to an effective pressure on surface (inward arrows: pressure) due to the surface tension of liquid gold, where color represents the magnitude of normalized displacement. A swelling is caused by surface tension at the upper central zone of each NP.

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
