# Peer review of "Plasmon-Enhanced Photothermal and Optomechanical Deformations of a Gold Nanoparticle"

_nanomaterials, 2020, doi:10.3390/nano10091881_

Round 1

Reviewer 1 Report

this paper starts from a basic misleading: the value of the surface tension of gold nanoparticles; there is a systematic difference between the evaluation of literature coming from different sources; the value considered by the authors is out of meaning; look at a recent review on this topic:

Beilstein J. Nanotechnol. 2018, 9, 2265–2276.

given this basic misleading in the value of surface tension all the paper is in my opinion not acceptable, unless completely revisioned

Author Response

Response: This suggested paper (Beilstein J. Nanotechnol. 2018, 9, 2265–2276) was useful and cited in our Reference. In this reference, the molecular dynamic (MD) simulation, based on reducing surface energy, was used to analyze the temperature-effect on the spatial distribution of liquid and solid phases in gold nanoparticle (NP). According to this paper’s conclusion, a quasi-liquid layer is first formed and then an outer liquid layer is induced on the whole surface of NP at the premelting stage due to the reduction of surface energy; their thicknesses gradually increase to the core as temperature rises. In fact, our model is consistent with this finding of MD model. Instead of using the concept of surface energy of MD simulation, we adopted a classical model of Young-Laplace, using an equivalent surface tension on the surface (effect of outer liquid layer) enclosed a semi-solid material inside (maxing state of quasi-liquid and solid phases), to analyze the elastic deformation of NP before fully melting.   

Fig. 9 in Beilstein J. Nanotechnol. 2018, 9, 2265–2276; a result of another Ref. [46]: J. Phys. Chem. C 2013, 117, 12289-12298.

Indeed, there is still a lot of room for improvement to the study of material’s properties at nanoscale, particularly when the temperature is close to the transition zone of melting point of NP. The MD simulation can be used to further study the transformation of solid-liquid phase. In the revised manuscript, we added several paragraphs in Abstract, Introduction and Method to mention the findings of molecular dynamics simulation based on the reduction of surface energy to support our assumption, a liquid layer covering the surface of NP, as follows.

“As the continuous heating process induces premelting of NPs, a quasi-liquid layer is formed first and then an outer liquid layer is induced due to the reduction of surface energy, predicted by the previous works of molecular dynamics simulation. Subsequently, we use the Young-Laplace model to investigate the surface tension effect on the following deformation.”

“In addition, we will also discuss the effect of quasi-liquid and liquid layers on the deformation of a premelting NP/NR, which is predicted from the molecular dynamic simulation based on the reduction of surface energy [35,36]. We will adopt a classical model of Young-Laplace, using an equivalent surface tension on the surface (the effect of outer liquid layer) enclosed a semi-solid material inside (an equivalent material of quasi-liquid and solid phases), to analyze the elastic deformation of NP before fully melting.”

“…Under continuous heating, the premelting of gold NP may take place. According to the estimate of molecular dynamic simulation, a quasi-liquid layer is formed first, and then an outer liquid layer covering the entire surface of NP to reduce surface energy is induced at the onset of melting; their thicknesses gradually increase and expand toward the core as temperature rises [35,36]. Once the outer liquid layer is produced, the surface-tension effect needs to be considered...”

Reviewer 2 Report

The paper is dedicated to a theoretical study of photothermal and optomechanical deformations in plasmon-enhanced gold nanoparticles.
I think that the subject matter is relevant to the nanomaterials journal, the work is well written and scientifically sound and the conclusions of NP cutting can be useful in photothermal optomechanics experiments.
Thus I support publication of the manuscript in the nanomaterials journal.

Author Response

Thanks.

Reviewer 3 Report

This manuscript reported the simulation of the laser-irradiation-induced stretching of a Au nanosphere that was experimentally reported by a previous work (ACS Nano 2019, 13, 32). The calculation can explain the experimental observation of the phenomenon that the shape of a Au nanosphere transforms from a sphere into an elongated particle and even a dimer under the laser irradiation. The reshaping was induced by the plasmon-enhanced photo-thermal conversion combined with optomechanical effects. The authors calculated the optical heating effects at the six different steps of the reshaping. Most importantly, the optical stretching strain distribution on the particle surface has been calculated carefully by the Maxwell Stress Tensor method. It is very interesting to have this work, which provides a theoretical guide for utilizing the photothermal conversion and optical force to tailor plasmonic gold nanostructures by focused laser beam. I therefore suggest the publication of this work on Nanomaterials if the authors could address the following comments properly.

  1. Could the author provide more explanation of how they deal with the adhesion effect of the molten Au nanoparticles on the substrate in the calculation? Should one consider the surface energy of the substrate in this study? The optical stretching process should be affected by the substrate, right?
  2. Could the authors provide the reference for the data of Young’s modulus and viscosity of Au in line 104?
  3. There are some typos and errors in the manuscript, for example, 1) the symbols are missing in line 88; 2) the format of the page number in reference 3 is wrong; 3) there is one more quotation mark in the title in reference 30.

Author Response

1. Could the author provide more explanation of how they deal with the adhesion effect of the molten Au nanoparticles on the substrate in the calculation? Should one consider the surface energy of the substrate in this study? The optical stretching process should be affected by the substrate, right?

Response: We didn’t consider the adhesion effect between nanoparticle and substrate and the surface energy in this work. This is because that Wang & Ding’s group modified Si substrate with a P(VDF-TrFE) layer to reduce the adhesion effect of gold NP on it for their experiment. However, the adhesion effect, depending on the surface condition of substrate, should affect the deformation of NP. We think that if no surface modification for substrate the optical stretching should be suppressed. To point out this effect, a sentence as follows was added in the Results and Discussion.

“….. Since Wang & Ding’s group modified Si substrate with a P(VDF-TrFE) layer for experiment, the adhesion effect between a molten gold NP and the surface-modified substrate was reduced [1]. Hence, we did not take into account the adhesion effect in our simulation. However, if the surface of substrate is not modified, the adhesion at the interface between gold NP and substrate may play an important role to suppress the optical-stretching deformation of the NP.”

2. Could the authors provide the reference for the data of Young’s modulus and viscosity of Au in line 104?

Response: According to Ref. of J. Nucl. Mater. 1967, 22, 28-32, the viscosity of gold at temperature of 600 oC is around 100 Pa×sec. Since the Young’s modulus of gold nanowire is about Es= 78 GPa (Nat. Materials 2005, 5, 525-9) and the value of liquid gold is EL @ 0, it is difficult to estimate the Young’s module E in the transition zone of melting point. In addition, the material is nonhomogeneous. Although it should be a function of temperature, we assume that the Young’s modulus E of gold NP of semi-solid at the premelting stage (say 590 C) is about E=  throughout this paper, where EL» 0.005 Pa; . In addition, for a soft material. To illustrate these parameters used for calculation, two references were added and a paragraph was added in the Method,

“…Since the Young’s modulus of gold nanowire is about Es= 78 GPa and the value of liquid gold is EL @ 0, it is difficult to estimate the Young’s module E in the transition zone of melting point [38]. Although it should be function of temperature, we assume that the Young’s modulus E of gold NP of semi-solid at the premelting stage (say 590 oC) is about E= = 2´104 Pa and throughout this paper, where EL» 0.005 Pa...”      

3. There are some typos and errors in the manuscript, for example, 1) the symbols are missing in line 88; 2) the format of the page number in reference 3 is wrong; 3) there is one more quotation mark in the title in reference 30.

Response: These typos and errors were corrected.

Round 2

Reviewer 1 Report

The paper has been greatly enhanced but there is a very impoortant point which remains in my opinion totally not sound and which is basic for the paper: the estimation of the surface energy of gold nanoparticles considered as high as 8j/m2; as indicated yet in a previous comment and as discussed in the ref. 36 these values are strongly suspected of a basic calculation error; the change of this value with the size of the nanoparticle should be in the opposite direction because the size reduction corresponds to reduction also of the surface tension,

Generally, considerations based on classical thermodynamics lead to the prediction of decreasing values of the surface energy with decreasing particle size. This phenomenon is caused by the reduced num- ber of next neighbors of surface atoms with decreasing particle size, a phenomenon that is partly compensated by the reduction of the binding energy between the atoms with decreasing particle size. Furthermore, this compensating effect may be expected by the formation of a disordered or quasi-liquid layer at the surface. The atomistic approach, based either on molecular dynamics simula- tions or ab initio calculations, generally leads to values with an opposite tendency. However, it is shown that this result is based on an insufficient definition of the particle size. A more realistic definition of the particle size is possible only by a detailed analysis of the electronic structure obtained from initio calculations. Except for minor variations caused by changes in the structure, only a minor dependence of the surface energy on the particle size is found

In my opinion the paper is not accpetbale untile this basic erros is not corrected

Reviewer 3 Report

The authors addressed the reviewers' comments properly. I therefore suggest the publication of this work as its current form.

Author Response

Thanks. The English style was checked. 

Round 3

Reviewer 1 Report

OK it is acceptable now